# How Depressing Is Inbreeding? A Meta-Analysis of 30 Years of Research on the Effects of Inbreeding in Livestock

**DOI:** 10.3390/genes12060926

**Published:** 2021-06-18

**Authors:** Harmen P. Doekes, Piter Bijma, Jack J. Windig

**Affiliations:** 1Animal Breeding and Genomics, Wageningen University & Research, 6700 AH Wageningen, The Netherlands; piter.bijma@wur.nl (P.B.); jack.windig@wur.nl (J.J.W.); 2Centre for Genetic Resources the Netherlands, Wageningen University & Research, 6700 AA Wageningen, The Netherlands

**Keywords:** homozygosity, cow, cattle, horse, pig, chicken, goat, sheep, rabbit

## Abstract

Inbreeding depression has been widely documented for livestock and other animal and plant populations. Inbreeding is generally expected to have a stronger unfavorable effect on fitness traits than on other traits. Traditionally, the degree of inbreeding depression in livestock has been estimated as the slope of the linear regression of phenotypic values on pedigree-based inbreeding coefficients. With the increasing availability of SNP-data, pedigree inbreeding can now be replaced by SNP-based measures. We performed a meta-analysis of 154 studies, published from 1990 to 2020 on seven livestock species, and compared the degree of inbreeding depression (1) across different trait groups, and (2) across different pedigree-based and SNP-based measures of inbreeding. Across all studies and traits, a 1% increase in pedigree inbreeding was associated with a median decrease in phenotypic value of 0.13% of a trait’s mean, or 0.59% of a trait’s standard deviation. Inbreeding had an unfavorable effect on all sorts of traits and there was no evidence for a stronger effect on primary fitness traits (e.g., reproduction/survival traits) than on other traits (e.g., production traits or morphological traits). *p*-values of inbreeding depression estimates were smaller for SNP-based inbreeding measures than for pedigree inbreeding, suggesting more power for SNP-based measures. There were no consistent differences in *p*-values for percentage of homozygous SNPs, inbreeding based on runs of homozygosity (ROH) or inbreeding based on a genomic relationship matrix. The number of studies that directly compares these different measures, however, is limited and comparisons are furthermore complicated by differences in scale and arbitrary definitions of particularly ROH-based inbreeding. To facilitate comparisons across studies in future, we provide the dataset with inbreeding depression estimates of 154 studies and stress the importance of always reporting detailed information (on traits, inbreeding coefficients, and models used) along with inbreeding depression estimates.

## 1. Introduction

Inbreeding depression refers to the decrease in mean phenotypic value with increased levels of inbreeding [1,2]. The phenomenon of inbreeding depression was already documented in the 19th century by Charles Darwin, who studied 57 plant species and observed that the offspring of self-fertilized plants were shorter, weighed less, flowered later, and produced fewer seeds than the offspring of unrelated plants [3,4]. Since Darwin’s time, inbreeding depression has been documented for a wide range of plant and animal species and for both wild and domestic populations [5,6,7,8,9,10].

Inbreeding depression is caused by an increase in homozygosity associated with inbreeding, which reduces the expression of dominance effects [1,2]. When dominance effects are on average favorable (i.e., when there is directional dominance in the favorable direction), the reduced expression of dominance effects results in a decrease in mean phenotypic value. In the absence of epistasis, the expected decrease in mean phenotypic value is linear and equals $-F\underset{i}{\sum }2p_{i}q_{i}d_{i}$, where F is the genome-wide inbreeding coefficient, $d_{i}$ is the dominance effect at locus i, and $p_{i}$ and $q_{i}$ are the allelic frequencies at locus i [1,2]. In the presence of epistasis, the relationship between mean phenotypic value and inbreeding may be nonlinear. Although deviations from linearity have been observed in some livestock populations (e.g., [11,12,13,14]), it is difficult to determine whether such deviations are truly the result of epistasis or are due to statistical artifacts [2]. Hence, a linear relationship between mean phenotypic value and inbreeding is commonly assumed.

The degree of inbreeding depression may differ across traits. Differences across traits may exist due to variation in the (relative) size of dominance effects, in the extent to which dominance effects act in the same direction or not, and in the role of epistasis. Meta-analyses among wild, zoo, and laboratory animal populations have suggested stronger inbreeding depression for primary fitness traits (e.g., fecundity, survival and development) than for morphometric traits (e.g., adult body size) and physiological traits (e.g., metabolic markers and parasite resistance) [2,6,7]. More recent analyses in wild and livestock populations, however, do not necessarily support this hypothesis [8,10].

Traditionally, the degree of inbreeding depression is quantified as the slope of the linear regression of phenotypes on pedigree-based inbreeding coefficients. With the increasing availability of genomic information, in particular single nucleotide polymorphism (SNP) data, pedigree inbreeding can be replaced by SNP-based inbreeding measures [15,16,17]. SNP-based measures include the percentage of homozygous SNPs [18], inbreeding derived from the diagonal of a genomic-relationship matrix (GRM) [19,20], and inbreeding based on runs of homozygosity (ROH) [21]. SNP-based measures may better predict homozygosity across the genome and, consequently, may better capture the negative consequences of homozygosity than pedigree inbreeding [22,23]. Among SNP-based measures, some may better estimate inbreeding depression than others. Various simulation studies have compared the use of different SNP-based measures for estimating inbreeding depression, with somewhat mixed results [22,24,25,26,27,28]. It would be valuable to summarize the findings from empirical studies comparing inbreeding depression estimates obtained from different pedigree-based and SNP-based measures of inbreeding.

In this study, we performed a meta-analysis of inbreeding depression estimates in livestock, based on 154 studies published from 1990 to 2020. Thereby, we extend the meta-analysis of Leroy [10], who evaluated 57 studies. Our objective was to assess and compare inbreeding depression estimates (1) across different trait groups, and (2) across different pedigree-based and SNP-based measures of inbreeding. In addition, we stress the importance of reporting detailed information (on traits, inbreeding measures, and models) along with inbreeding depression estimates to facilitate meta-analyses in future.

## 2. Materials and Methods

### 2.1. Literature Search

A literature search was performed in Web of Science [29] and Scopus [30] on December 29, 2020. Species included were cattle, pig, chicken, sheep, goat, horse, and rabbit (as in Leroy [10]). The search phrase was *(“inbreeding depression” OR “effect* of inbreeding”) AND (cattle OR cow* OR bull* OR pig* OR chicken* OR sheep* OR goat* OR horse* OR rabbit*)*. In total, 696 hits from Web of Science and 532 hits from Scopus were obtained. After merging these hits and removing duplicates, 766 unique studies remained.

Further filtering was performed with the aim to identify studies that reported inbreeding depression as linear regression coefficients (b-values). Studies were discarded when (i) they were published before 1990; (ii) they were about non-target species (e.g., guinea pigs, horseshoe bats, or rabbiteye blueberries); (iii) they were about non-livestock populations (e.g., wild or zoo populations); (iv) they did not report trait means nor standard deviations; (v) they did not report b-values, but effects of inbreeding classes; (vi) they used quadratic or exponential regression models, in which the quadratic/exponential effects were significant; (vii) it was unclear if the effects were reported per 1%, 10%, or 100% increase in inbreeding; or (viii) when the full article was not available through the library services of Wageningen University & Research. After filtering using these criteria, a total of 143 studies remained. An additional eight studies, which were included in the meta-analyses of Leroy [10] or Bezdíček [31] and were not found with the above search strategy, were added to the dataset. Another three studies that were published at the beginning of 2021 were also added. The final dataset included 154 studies [11,12,32,33,34,35,36,37,38,39,40,41,42,43,44,45,46,47,48,49,50,51,52,53,54,55,56,57,58,59,60,61,62,63,64,65,66,67,68,69,70,71,72,73,74,75,76,77,78,79,80,81,82,83,84,85,86,87,88,89,90,91,92,93,94,95,96,97,98,99,100,101,102,103,104,105,106,107,108,109,110,111,112,113,114,115,116,117,118,119,120,121,122,123,124,125,126,127,128,129,130,131,132,133,134,135,136,137,138,139,140,141,142,143,144,145,146,147,148,149,150,151,152,153,154,155,156,157,158,159,160,161,162,163,164,165,166,167,168,169,170,171,172,173,174,175,176,177,178,179,180,181,182,183].

### 2.2. Inbreeding Depression Estimates and Trait Classification

A total of 2321 inbreeding depression estimates were retrieved from the 154 studies. Studies typically reported multiple estimates for different traits, for different breeds, for direct and parental inbreeding effects and/or for different inbreeding measures. The number of estimates per study ranged from 1 to 436, with a mean of 15.2 and a median of 6. The complete dataset is available in the Appendix A.

To reduce the number of unique traits in the analyses, similar traits were combined into a single trait. For example, traits like calving interval, days open, and the interval between calving and insemination were combined into a single trait “fertility interval”. Traits were classified into six trait groups: reproduction/survival, weight/growth, production, conformation, health, and other traits. Classification into trait groups was similar as in Leroy [10], plus an extra group of health traits, which included somatic cell score (SCS), disease traits, and locomotion.

For each trait, the favorable phenotypic direction was determined. For traits where the favorable direction was an increase in mean phenotypic value (e.g., milk yield or body size), the b-values were used with their original sign. For traits where the favorable direction was a decrease (e.g., SCS or mortality), the sign of b-values was changed. Traits with an optimum or an unclear favorable direction (e.g., foot angle or meat pH) were excluded from the analyses (*n* = 197 estimates, so that 2124 estimates remained).

### 2.3. Comparison across Traits and Trait Groups

To enable comparison across traits and trait groups, b-values were scaled by dividing them by the trait mean (to obtain $b_{m}$) or trait standard deviation (SD; to obtain $b_{s}$). Not every study reported trait means and SDs. Consequently, the number of available estimates for $b_{m}$ and $b_{s}$ equaled 2094 and 1519, respectively. For the comparison across traits and trait groups, SNP-based estimates were excluded (*n* = 257 for $b_{m}$ and *n* = 255 for $b_{s}$). In addition, outliers that were more than 3 SDs away from the mean were excluded (*n* = 19 for $b_{m}$ and *n* = 5 for $b_{s}$). After these edits, 1818 and 1259 pedigree-based estimates remained for $b_{m}$ and $b_{s}$, respectively. Descriptive statistics for $b_{m}$ and $b_{s}$ were calculated in R [184]. Skewness and kurtosis were determined with the package “moments” [185]. The following model was then applied:$$Y_{ijk}=\mu +POPULATION_{i}+TRAIT\text{\_}GROUP_{j}+E_{ijk}$$
where $Y_{ijk}$ was the inbreeding depression estimate (either $b_{m}$ or $b_{s}$); $POPULATION_{i}$ was the effect of the *ith* population; $TRAIT\text{\_}GROUP_{j}$ was the effect of the *jth* trait group; and $E_{ijk}$ was the error term. Population was defined as the combination of study and breed. Study and breed were combined into a single effect, because many studies focused on a single breed, and breeds were often investigated in a single study. The model was run with the “glm” function in R [184]. Estimated marginal means (EMMs), also known as least square means, were obtained for the different trait groups with the function “emmeans” from the package “emmeans” [186]. Pairwise comparisons between EMMs were performed with the function “pwpm” in emmeans [186], which applies Tukey–Kramer’s procedure to account for multiple testing [187,188]. To study potential differences between individual traits, the same model as above was used, but with $TRAIT_{j}$ instead of $TRAIT\text{\_}GROUP_{j}$. In the latter analysis, traits with less than 10 records were excluded.

### 2.4. Comparison across Inbreeding Measures

For the comparison across inbreeding measures, the following measures were considered: pedigree inbreeding ($F_{PED}$), inbreeding based on ROH ($F_{ROH}$), inbreeding derived from the diagonal of a GRM ($F_{GRM}$), inbreeding derived from the diagonal of a GRM computed with allele frequencies of 0.5 ($F_{GRM0.5}$), and the percentage of homozygous SNPs (HOM). Note that $F_{GRM0.5}$ and HOM are equivalent, except for a difference in scale, with $F_{GRM0.5}=2HOM-1$ (Appendix B). Measures based on GRMs were combined into $F_{GRM}$, no matter which GRM was used. Measures based on ROH were considered as $F_{ROH}$, regardless of the criteria used to identify ROH (which can vary substantially, e.g., [189]).

Comparisons were made within studies and within traits. There were 12 studies that reported b-values for at least two of the above-mentioned inbreeding measures [129,134,138,146,155,157,160,168,173,177,181,183]. These studies all reported trait means and SDs, which were used to calculate $b_{m}$ and $b_{s}$. The dataset used for the comparisons is provided in the Appendix A. A direct comparison of $b_{m}$ and $b_{s}$ is inappropriate, because of scale differences between $F_{PED}$, $F_{ROH}$, $F_{GRM}$, $F_{GRM0.5}$, and HOM [26]. For example, HOM measures the probability of alleles being “identical by state” (IBS) and typically has a mean of 60 to 70% and a SD of 1 to 2%, whereas $F_{PED}$ measures the probability of alleles being “identical by descent” (IBD) with reference to the founder generation and typically has a lower mean and larger SD than HOM (Appendix A). To account for such differences, $b_{m}$ and $b_{s}$ were standardized by multiplying them by the SD of the inbreeding measure in the corresponding population, so that they were expressed per 1 SD increase in inbreeding (rather than per 1%). For one of the twelve studies [134], scaling was not possible because SDs of inbreeding measures were not reported. This study was therefore excluded from the comparison across inbreeding measures.

Significance of b-values was also compared across inbreeding measures. Test statistics were calculated as ${\left(\frac{b}{se\left(b\right)}\right)}^{2}$ and these test-statistics were compared to a chi-square distribution with one degree of freedom to obtain approximate *p*-values (following the Wald test). Smaller *p*-values indicate a more significant association between inbreeding and phenotypic value and suggest more predictive power to detect inbreeding depression.

## 3. Results

### 3.1. Inbreeding Depression Estimates for Different Traits and Trait Groups

Across all studies and traits, the median (mean) pedigree-based $b_{m}$ and $b_{s}$ equaled −0.13 (−0.22) and −0.59 (−0.71), respectively. In other words, a 1% increase in $F_{PED}$ was associated with a median decrease in phenotypic value of 0.13% of a trait’s mean, or 0.59% of a trait’s SD. The distributions of $b_{m}$ and $b_{s}$ showed substantial kurtosis (i.e., were heavily tailed) and were somewhat negatively skewed (Figure 1).

For each trait group, except for the group “other traits”, the mean and median $b_{m}$ and $b_{s}$ were negative (Figure 2). After correcting for the effect of population, the EMMs of $b_{m}$ and $b_{s}$ for all groups, except “other traits”, were all negative and most were significantly below zero (Table 1). The EMMs for $b_{m}$ and $b_{s}$ for “other traits” were positive but not significantly different from zero (*P* > 0.05). When ignoring the “other traits” and comparing groups based on $b_{m}$, production traits and reproduction/survival traits showed the most depression (EMMs of −0.308 and −0.302, respectively) and conformation traits showed the least depression (EMM of −0.142). The difference between production and reproduction/survival traits on one hand, and conformation traits on the other hand, was also significant (*P* < 0.05), whereas other pairwise comparisons were not (Appendix A). When comparing trait groups based on $b_{s}$, weight/growth traits showed the most depression (EMM of −1.071) and reproduction/survival traits showed the least depression (EMM of −0.410). For $b_{s}$, there were no significant pairwise comparisons between EMMs of trait groups (except with the group “other traits”; Appendix A).

When running a model with individual traits instead of trait groups, the EMMs for individual traits showed substantial variation (Appendix A). For $b_{m}$, the EMMs ranged from −0.938 to 0.321, with the vast majority below zero (39 out of 44, of which 19 with *P* < 0.05). The EMMs above zero were not significant (*P* > 0.05). For $b_{s}$, the EMMs ranged from −1.938 to 1.522, with the majority being below zero (26 out of 29, of which there were eight with *P* < 0.05) and only one EMM was significantly larger than zero (*P* < 0.05).

### 3.2. Inbreeding Depression Estimates for Pedigree-Based and SNP-Based Measures of Inbreeding

Standardized $b_{m}$ and $b_{s}$ generally correlated well across different inbreeding measures (Figure 3 and Figure 4). The highest correlations were found between $F_{GRM0.5}$ and HO (0.97 for $b_{m}$ and 0.94 for $b_{s}$). Correlations between $F_{ROH}$ and HOM, and correlations between $F_{GRM}$ and $F_{GRM0.5}$, were also high (i.e., ≥0.9). The lowest correlations were found between $F_{ROH}$ and $F_{GRM}$ (0.27 for $b_{m}$ and 0.52 for $b_{s}$) and between $F_{GRM}$ and HOM (0.24 for $b_{m}$ and 0.47 for $b_{s}$), although it should be noted that these correlations were low due to a single study (shown in purple in Figure 3 and Figure 4).

When comparing the significance of inbreeding depression estimates, SNP-based inbreeding measures generally had lower *p*-values than pedigree inbreeding (Table 2). For example, $F_{ROH}$ had a lower *p*-value than $F_{PED}$ in 30 out of 38 comparisons (79%). In these comparisons, the median *p*-value was 0.026 for $F_{ROH}$ and 0.131 for $F_{PED}$. Similarly, the *p*-values for $F_{GRM}$, $F_{GRM0.5}$ and HOM were lower than those for $F_{PED}$ in the majority of comparisons (7 out of 7 for $F_{GRM}$, 21 out of 26 for $F_{GRM0.5}$, and 15 out of 18 for HOM).

Among the SNP-based measures, no consistent differences in *p*-values were observed (Table 2). For example, the percentage of comparisons in which the second measure had a lower *p*-value than the first measure was 57% for the comparisons between $F_{ROH}$ and $F_{GRM}$, 56% between $F_{ROH}$ and HOM, and 50% between $F_{GRM0.5}$ and HOM. The latter was expected because of the equivalence of $F_{GRM0.5}$ and HOM (Appendix B). In the comparison between $F_{ROH}$ and $F_{GRM0.5}$, $F_{ROH}$ had a lower *p*-value in 25 out of 40 comparisons (62%), but the median *p*-value was very similar for the two measures (0.037 vs. 0.036). In the comparison between $F_{GRM}$ and HOM, $F_{GRM}$ had a lower *p*-value in 10 out of 14 comparisons (71%), whereas the median *p*-value was smaller for HOM than for $F_{GRM}$ (0.009 vs. 0.071). In the comparison between $F_{GRM}$ and $F_{GRM0.5}$, $F_{GRM0.5}$ had a lower *p*-value in 100% of the comparisons, but this was based on only three comparisons in a single study.

## 4. Discussion

In this meta-analysis, 154 studies were evaluated. The objective was to assess and compare the degree of inbreeding depression across different trait groups and across different pedigree-based and SNP-based measures of inbreeding.

Across all studies and traits, a 1% increase in pedigree inbreeding was associated with a median (mean) decrease in phenotypic value of 0.13% (0.22%) of a trait’s mean, or 0.59% (0.71%) of a trait’s standard deviation. These effects are similar to the mean $b_{m}$ of 0.14% and mean $b_{s}$ of 0.56% reported by Leroy [10]. Distributions of $b_{m}$ and $b_{s}$ showed substantial kurtosis and were negatively skewed (Figure 1). The observed kurtosis might be the result of the final distribution being a mixture of underlying distributions with the same mean, but different SD (where studies with small sample size have a larger SD of estimates). Such a mixture can have a higher kurtosis than the separate distributions, as illustrated in Figure 5. The observed negative skewness could be due to publication bias [7,8,190]. Multiple studies explicitly stated that non-significant estimates were not reported (e.g., [93,130,147,178]). Omitting non-significant results does not necessarily introduce bias, as long as results in both directions are equally likely to be omitted. To further investigate the presence of publication bias, we retrieved the number of records per study (when reported) and made a funnel plot with $b_{m}$ on the *x*-axis and the number of records on the *y*-axis (Figure 6). As expected, the funnel plot showed more variation in $b_{m}$-estimates for studies with few records compared to studies with many records. In addition, for studies with relatively few records, positive inbreeding effects were relatively scarce compared to negative outliers, suggesting indeed some degree of publication bias.

Across trait groups, there were some differences in mean $b_{m}$ and $b_{s}$ (Table 1, Appendix A). These differences, however, were not consistent for $b_{m}$ and $b_{s}$ and did not support the hypothesis that primary fitness traits such as survival and reproduction exhibit more inbreeding depression than other traits. In fact, when comparing the EMMs of $b_{s}$ across trait groups, reproduction/survival traits showed the least inbreeding depression (except for the group “other traits”). This is similar to Leroy [10], who reported EMMs for $b_{m}$ and $b_{s}$ of −0.222 and −0.336 for reproduction/survival, of −0.092 and −0.473 for conformation, of −0.24 and −0.563 for weight/growth, of −0.351 and −0.817 for production, and of −0.093 and −0.488 for other traits. The relatively mild EMM of $b_{s}$ for reproduction/survival traits could be the result of such traits generally showing more phenotypic variation due to environmental sources, which is in analogy with the observation that reproduction/survival traits generally have lower heritabilities than other traits, whereas coefficients of genetic variation (also known as “evolvabilities”) tend to be more similar across trait groups [191].

The hypothesis that fitness traits exhibit more inbreeding depression is largely based on results from wildlife and laboratory populations. In a survey among laboratory populations of *Drosophila melanogaster*, Lynch and Walsh [2] observed a high degree of inbreeding depression for primary fitness traits such as viability, fertility and egg production, and a low degree of inbreeding depression for morphological traits. In a meta-analysis of non-domestic animal populations, DeRose and Roff [6] also reported more inbreeding depression for life history traits (fecundity, survival and development) than for morphological traits (adult body size). They reported a median $b_{m}$ and $b_{s}$ of −0.47 and −1.45 for life history traits and of −0.09 and −0.59 for morphological traits. Coltman and Slate [7] performed a meta-analysis on correlations between phenotypes and two measures of genetic variation at microsatellite loci, multilocus heterozygosity (MLH), and mean squared allele size differences ($d^{2}$). Using data from domestic and non-domestic populations, they found significant correlations for life history traits (0.0856 for MLH and 0.0479 for $d^{2}$) and smaller non-significant correlations for morphometric traits (0.0052 for MLH and 0.0038 for $d^{2}$) and physiological traits (0.0075 for MLH and 0.0055 for $d^{2}$). It should be noted that these estimates were obtained while not accounting for dependence between estimates from the same studies and the same populations within studies. When they analyzed the average per trait group within study units (“study unit average” approach), the difference between correlations for the different trait groups decreased and confidence intervals overlapped [7]. Chapman et al. [8] found similar results. In their meta-analysis of heterozygosity-fitness correlations, they also found that the study unit average approach resulted in smaller differences in correlations across trait groups than when they treated all records independently. They additionally used a linear mixed model to account for a population effect, which resulted in even more similar correlations across trait groups than the study unit average approach, with confidence intervals that strongly overlapped. With the mixed model, they found a mean correlation (and confidence interval) between phenotype and MLH of 0.098 (0.0674–0.1293) for life history traits, of 0.0611 (0.0302‒0.0919) for morphometric traits, and of 0.0809 (0.0048–0.1560) for physiological traits [8]. Thus, although the empirical results are equivocal, there are indications from wild and laboratory populations that fitness traits might exhibit more inbreeding depression.

There are also theoretical arguments why primary fitness traits would show more inbreeding depression than traits less related to fitness. Inbreeding depression depends on directional dominance, as indicated by the expected inbreeding depression that (in absence of epistasis) equals $-F\underset{i}{\sum }2p_{i}q_{i}d_{i}$ [1,2]. To exhibit inbreeding depression, a trait should be influenced by dominance effects (resulting in dominance variance) and, more importantly, these dominance effects should be favorable on average. For traits strongly under directional selection such as a primary fitness trait like survival, the average dominance effect is expected to be favorable, because fixation occurs more quickly for loci with an unfavorable dominance effect [2]. For a trait that is less related to fitness as well as for traits under stabilizing selection, directional dominance is expected to be less pronounced because of the lower directional selection pressure.

Given that directional dominance is a function of how much a trait has been selected upon, the results of this meta-analysis (and that of Leroy [10]) are in line with expectation. Livestock populations are typically under directional selection for a combination of production, conformation, growth, reproduction, survival, behavioral and health traits (in addition to natural selection on primary fitness traits). Hence, all of these trait groups may show a similar degree of directional dominance and, consequently, a similar degree of inbreeding depression. This is also in line with the relative dominance variance (i.e., the proportion of dominance variance over phenotypic variance), which appears to be similar across trait groups in livestock. For example, Doekes et al. [192] compared estimates of relative dominance variance across five studies in cattle and found no clear differences between yield, fertility, and health traits. Thus, any breeding goal trait in livestock can be considered under selection just as a fitness trait and may exhibit considerable inbreeding depression.

The efficiency of selection against (partially) deleterious alleles is increased by inbreeding, a process called purging [193,194]. Since livestock populations typically have relatively small effective population sizes (although they may have very large census sizes), purging can be efficient to reduce the inbreeding load in these populations [194]. Since purging acts on all traits under selection, it is not expected to cause differences in the degree of inbreeding depression between traits under selection. Nevertheless, the effective population size influences purging [193,194] and it would be interesting to study the association between effective population size and observed inbreeding depression.

In this meta-analysis, we compared inbreeding depression estimates across different pedigree-based and SNP-based measures by standardizing $b_{m}$ and $b_{s}$ (expressing them per 1 SD increase in the inbreeding measure) and by comparing *p*-values based on reported b-values and corresponding standard errors. Alternatively, it has been suggested to compare correlations between inbreeding measures and phenotype [26]. The use of correlations, however, was unfeasible in this meta-analysis, because most livestock studies report inbreeding depression as b-values obtained from animal models.

Standardized $b_{m}$- and $b_{s}$-estimates correlated well across the different pedigree-based and SNP-based inbreeding measures (Figure 3 and Figure 4). These correlations are expected to be largely driven by the correlations between the underlying inbreeding estimators and, with that in mind, they also followed expectation. For example, the high correlations between inbreeding depression estimates for HOM and $F_{ROH}$ (0.96 for $b_{m}$ and 0.92 for $b_{s}$) are in line with the high reported correlations between the coefficients of HOM and $F_{ROH}$ in the underlying studies (e.g., 0.81 [129], 0.94 [157], and 0.86 [181]). Similarly, the moderate correlations between inbreeding depression estimates for $F_{PED}$ and $F_{ROH}$ (0.52 for $b_{m}$ and 0.65 for $b_{s}$) are in line with the moderate reported correlations between the coefficients of $F_{PED}$ and $F_{ROH}$ in the underlying studies (of e.g., 0.66 [168], 0.63 [177], and 0.60 [181]). *p*-values of inbreeding depression estimates were smaller for SNP-based inbreeding measures than for pedigree inbreeding (Table 2). SNP-based measures may be more accurate than pedigree inbreeding because the former account for Mendelian sampling (e.g., [195]) and do not depend on pedigree completeness and quality (e.g., [196]). Since measurement errors in the independent variable lead to downward bias in the estimated slope (a statistical phenomenon called “regression dilution”), regression on less accurate pedigree-based coefficients may result in smaller b-values than regression on more accurate SNP-based coefficients. The benefit of using SNP-based measures will, among others, depend on the number of SNPs in relation to the genome length and effective population size (e.g., [23]). Having more SNPs available is expected to allow for a better estimation of the realized inbreeding and, therefore, of the realized inbreeding depression. In addition, when SNP-based measures depend on allele frequencies in the population (e.g., $F_{GRM}$), a sufficient number of individuals is required to accurately estimate allele frequencies.

Since the scale of inbreeding measures strongly influences b-values, only pedigree-based estimates were used for the comparison between trait groups. One may argue that the pedigree-based comparison of b-values across studies is also inappropriate because the scale of $F_{PED}$ differs across populations (due to differences in pedigree depth). To account for differences in scale of $F_{PED}$, we initially considered the SD of $F_{PED}$ as an explanatory variable in our model. However, since this SD was available for less than a third of the estimates, and since its effect was not significant in the preliminary analyses, it was removed from the final model (as presented in the Materials and Methods). A population effect, defined as breed within study, was included in the model, which indirectly may have accounted for the population-specific SD of inbreeding coefficients.

No clear differences in effect sizes and *p*-values were found between inbreeding depression estimates for $F_{ROH}$, $F_{GRM}$, $F_{GRM0.5},$ and HOM (Figure 3 and Figure 4 and Table 2). However, it is difficult to draw firm conclusions, because of (1) the limited number of empirical studies directly comparing the different measures; (2) the arbitrary definitions of especially $F_{ROH}$, with many different criteria that are often not fully reported [189]; and (3) not knowing the true inbreeding depression. These limitations can be partly overcome in simulation studies. Various simulations studies have been performed to investigate which SNP-based measure might be most appropriate to estimate inbreeding depression. Keller et al. [22] investigated the correlation between different inbreeding measures and the homozygous mutation load (HML), which they defined as the number of homozygous loci for rare alleles (with MAF < 0.5) in an individual. They found that $F_{ROH}$ had more power to detect HML than $F_{PED}$ or excess of SNP-by-SNP homozygosity. Kardos et al. [24] found that SNP-based measures (using 35k SNPs) better explained the variation in realized genomic IBD than $F_{PED}$ (with 20 generations known). They also found that $F_{ROH}$ and the excess of SNP-by-SNP homozygosity explained very similar amounts of variation in realized IBD. Yengo et al. [25] used the SNP data of humans and simulated inbreeding depression by assigning phenotypic effects to samples of SNPs. They reported that $F_{GRM}$ from Yang’s method [20] performed best. This is a SNP-by-SNP measure with high weight on rare alleles that is also known as $F_{UNI}$ (“inbreeding based on the correlation of uniting gametes”). In their study, $F_{ROH}$ resulted in the overestimation of inbreeding depression. In the study of Nietlisbach et al. [27], in contrast, $F_{ROH}$ provided unbiased results, whereas $F_{GRM}$ from Yang’s method [20] resulted in upwardly biased inbreeding depression estimates. Recently, Caballero et al. [28] showed that these (seemingly) contradictory results might be explained by population characteristics. They found that inbreeding depression estimates obtained from $F_{ROH}$ were appropriate for populations with small effective sizes (e.g., $N_{e}$ = 100), but were downwardly biased for populations with large effective sizes (e.g., $N_{e}$ = 5000), unless sufficiently long ROHs (>5 Mb) were used. Inbreeding depression estimates based on $F_{GRM}$ from Yang’s method [20], on the other hand, were upwardly biased for populations with small effective sizes, but were nearly unbiased for populations with large effective sizes. They also studied $F_{GRM}$ from VanRaden’s method 2 [19] and $F_{HOM}$, the latter being a measure of the deviation from Hardy–Weinberg frequencies that has a correlation of 1 with HOM and $F_{GRM0.5}$ when base allele frequencies are known. These measures always underestimated inbreeding depression. Overall, Caballero et al. [28] concluded that it depends on the population as to which SNP-based inbreeding measure is most appropriate to estimate inbreeding depression.

Aside from providing more powerful measures to detect inbreeding depression, genomic data offer additional opportunities to study the genetic background of inbreeding depression. First, the length of ROHs can be used to study the effect of the age of inbreeding on inbreeding depression, in addition to already existing pedigree-based methods [168,177]. Recent inbreeding may be more harmful than ancient inbreeding, as a result of purging. Using genomic time series, the process of purging could also be studied in more detail. Second, SNP-data can be used to search for genomic regions associated with inbreeding depression (e.g., through ROH scans [192,197,198]). It should be noted, however, that such methods are prone to statistical issues such as multiple testing and tend to go against the infinitesimal model (i.e., the idea that inbreeding depression is predominantly caused through many loci with small effects). Third, with genomic data, it has become possible to study the role of regulatory mechanisms such as methylation, in explaining inbreeding depression [199]. Last, genomic data may help to shed more light on the role of partial dominance, overdominance, and epistasis in explaining inbreeding depression.

Various studies were excluded from this meta-analysis, because they lacked detailed information on the inbreeding depression estimates. Here, we therefore list recommendations on how to report inbreeding depression estimates. First, it is important to report descriptive statistics of the traits that are investigated (number of animals, mean, SD, etc.). This allows future studies, among others, to scale b-values (e.g., to $b_{m}$ and $b_{s}$) and to investigate the effect of population size on the results (as e.g., in Figure 6). The descriptive statistics should be provided for the individuals used in the analysis and, when multiple populations or subgroups (e.g., breeds, sexes, or age classes) are studied, statistics should be reported for each population separately. Second, it is important to provide details on how inbreeding measures were calculated. For $F_{PED}$, for example, this includes information on pedigree completeness such as the complete generation equivalent or number of complete generations. Again, such statistics should be reported for the individuals used in the analysis and not (only) for the entire pedigree. For $F_{ROH}$, the criteria and approach to identify ROHs should be fully explained. Third, descriptive statistics on the inbreeding measures (distribution, mean, SD) should be provided to enable scaling (as e.g., in Figure 3 and Figure 4). Fourth, when estimating b-values, it is important to correct for appropriate fixed effects as well as for additive genetic effects [200] (e.g., with an animal model). Fifth, it should be clearly stated whether b-values are expressed per 1%, 10%, or 100% increase in inbreeding. Sixth, for traits such as litter size, calving ease, and success of insemination, it should be clearly stated whether b-values correspond to a regression of phenotypes on inbreeding of the offspring/litter, on inbreeding of the dam (‘maternal’), or on inbreeding of the sire (‘paternal’). Seventh, b-values should be reported with standard errors and sufficient decimal places to facilitate calculation of test statistics. Eighth, it is important to also report non-significant and/or favorable effects to prevent publication bias. Last, it would be valuable to provide estimates of the effective population size, when available, because the effective population size influences the amount of genetic purging in the population and, thus, the degree of inbreeding depression. Overall, these recommendations should contribute to enable comparisons of inbreeding depression estimates across studies and facilitate meta-analyses in future.

## 5. Conclusions

Inbreeding has an unfavorable effect on livestock traits. Based on this meta-analysis, a 1% increase in pedigree inbreeding is associated with a median decrease in phenotypic value of 0.13% of a trait’s mean, and 0.59% of a trait’s SD. Various trait groups (i.e., reproduction/survival, weight/growth, conformation, production, and health) show a similar degree of inbreeding depression. *p*-values of inbreeding depression estimates for SNP-based inbreeding measures were smaller than those for pedigree inbreeding, suggesting more power for SNP-based measures. There were no consistent differences in *p*-values for percentage of homozygous SNPs, inbreeding based on ROH or inbreeding based on a genomic relationship matrix. Comparisons between measures, however, are difficult because of the limited number of studies that directly compares them, the different scales of measures, and arbitrary definitions for particularly ROH-based inbreeding. To facilitate comparisons across studies in future, we highly recommend always reporting detailed information about inbreeding depression estimates (on traits, inbreeding coefficients, and models used).

## Figures and Tables

**Figure 1 genes-12-00926-f001:**
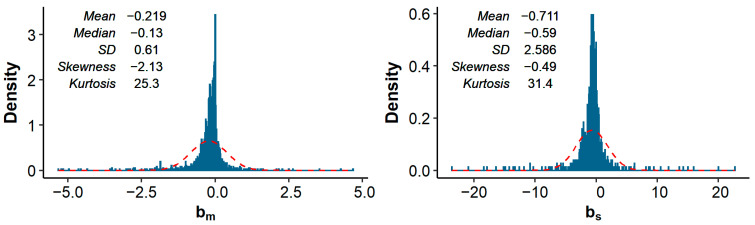
Histograms of estimates of $b_{m}$ (*n* = 1818) and $b_{s}$ (*n* = 1259) across all studies and traits, after removal of extreme outliers. Descriptive statistics and a normal distribution (dashed red lines; based on mean and standard deviation (SD)) are also shown.

**Figure 2 genes-12-00926-f002:**
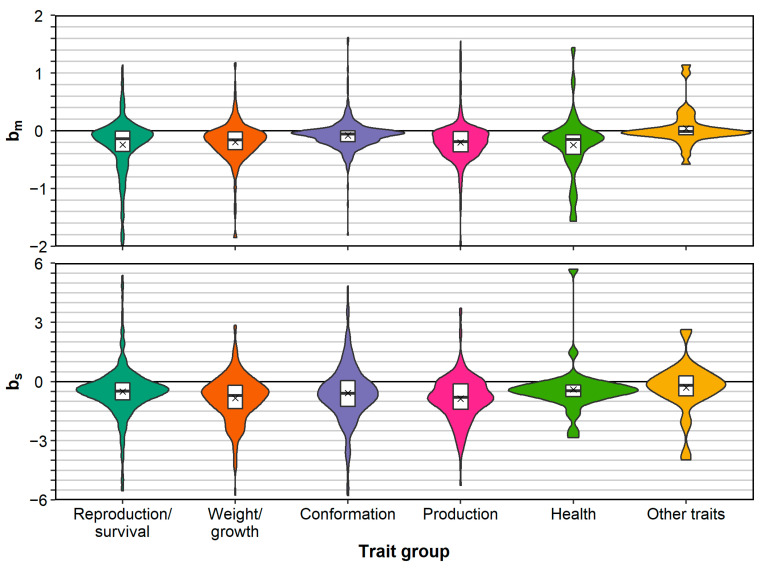
Violin plots of inbreeding depression estimates per trait group. Estimates are expressed as a percentage of a trait’s mean ($b_{m}$) or as a percentage of a trait’s SD ($b_{s}$). Boxplots are also shown, indicating the median, 25th and 75th quantiles and the mean (×) for each group. For $b_{m}$ and $b_{s}$, there were respectively 40 and 39 extreme estimates outside the range of this figure.

**Figure 3 genes-12-00926-f003:**
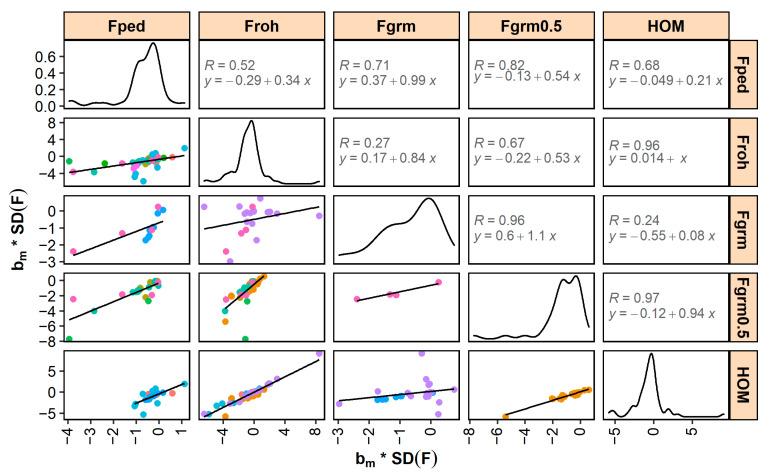
Relationship between inbreeding depression estimates expressed as percentage of a trait’s mean per 1 standard deviation increase in inbreeding ($b_{m}$ * SD(F)) across different measures of inbreeding. The data points (colored per study) and linear trendline are shown (lower triangle) as well as the density curve for each inbreeding measure (diagonal) and the correlation and regression equation (upper triangle). Note that slopes of the linear trendline differ from 1, which is also expected when correlations between inbreeding measures themselves are not equal to 1. $F_{PED}$ = pedigree inbreeding; $F_{ROH}$ = inbreeding based on runs of homozygosity; $F_{GRM}$ = inbreeding from genomic relationship matrix (studies in pink and purple used VanRaden’s method 2, and light blue Yang’s method); $F_{GRM0.5}$ = inbreeding from genomic relationship matrix with allele frequencies fixed to 0.5; HOM = percentage of homozygous SNPs.

**Figure 4 genes-12-00926-f004:**
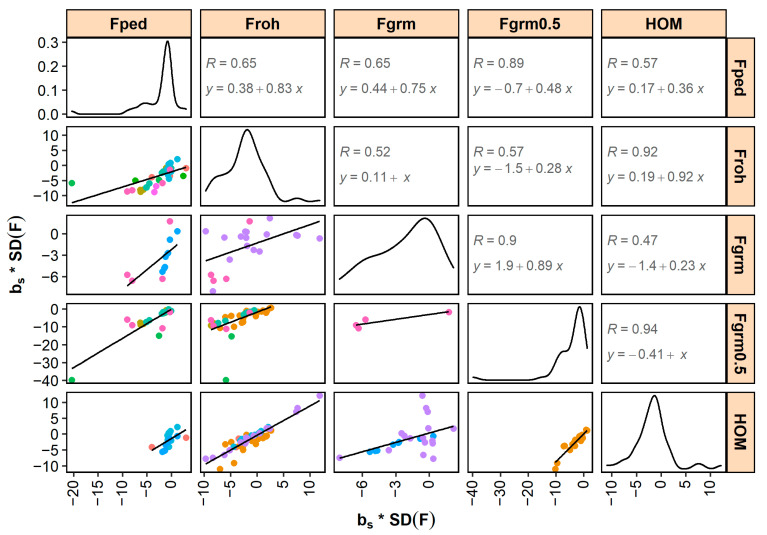
Relationship between inbreeding depression estimates expressed as percentage of a trait’s standard deviation per 1 standard deviation increase in inbreeding ($b_{s}$ * SD(F)) across different measures of inbreeding. The data points (colored per study) and linear trendline are shown (lower triangle), as well as the density curve for each inbreeding measure (diagonal) and the correlation and regression equation (upper triangle). Note that slopes of the linear trendline differ from 1, which is also expected when correlations between inbreeding measures themselves are not equal to 1. $F_{PED}$ = pedigree inbreeding; $F_{ROH}$ = inbreeding based on runs of homozygosity; $F_{GRM}$ = inbreeding from genomic relationship matrix (studies in pink and purple used VanRaden’s method 2, and light blue used Yang’s method); $F_{GRM0.5}$ = inbreeding from genomic relationship matrix with allele frequencies fixed to 0.5; HOM = percentage of homozygous SNPs.

**Figure 5 genes-12-00926-f005:**
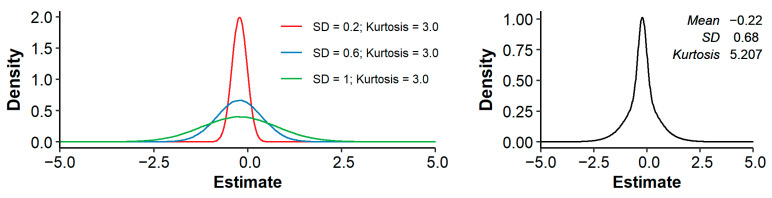
Example with three normal distributions (**left**; each with a mean of −0.22 and a SD of 0.2, 0.6 or 1) and the resulting mixture of these three normal distributions (**right**), showing an increase in kurtosis.

**Figure 6 genes-12-00926-f006:**
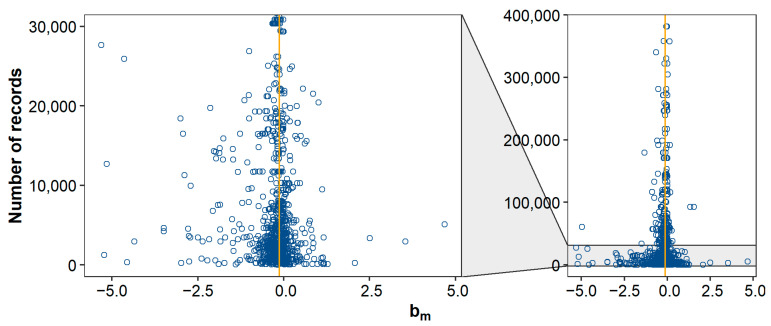
Funnel plot to assess publication bias. The plot shows the relationship between inbreeding depression estimates, expressed as a percentage of the trait mean ($b_{m}$), and the number of records used to estimate them (N = 1283). The orange vertical line represents the median. To ease interpretation, estimates based on >400,000 records are not shown (N = 82).

**Table 1 genes-12-00926-t001:** Estimated marginal means of inbreeding depression estimates per trait group with standard errors (SEs). Inbreeding depression estimates are expressed as percentage of a trait’s mean ($b_{m}$) or as percentage of a trait’s SD ($b_{s}$). The number of estimates (N) and the *p*-value for testing the mean against zero are also shown.

		$b_{m}$			$b_{s}$	
Trait Group	N	Estimate (SE)	*p*-Value	N	Estimate (SE)	*p*-Value
Reproduction/survival	590	−0.302 (0.032)	<0.001	349	−0.410 (0.182)	0.024
Weight/growth	417	−0.227 (0.039)	<0.001	231	−1.071 (0.244)	<0.001
Conformation	419	−0.142 (0.046)	0.002	396	−0.487 (0.210)	0.020
Production	319	−0.308 (0.040)	<0.001	216	−0.753 (0.215)	<0.001
Health	39	−0.268 (0.099)	0.007	35	−0.891 (0.464)	0.055
Other traits	34	0.129 (0.103)	0.213	32	0.826 (0.471)	0.079

**Table 2 genes-12-00926-t002:** Comparison of *p*-values of inbreeding depression estimates based on different pedigree-based and SNP-based inbreeding measures. Comparisons were made within studies and for each combination of two inbreeding measures, where both inbreeding measures had an unfavorable effect on the phenotype.

						Comparisons *P*2 < *P*1	
Measure 1	Measure 2	N Studies	N Comparisons	Median *P*1	Median *P*2	N	% of Total
$F_{PED}$	$F_{ROH}$	8	38	0.131	0.026	30	79
$F_{PED}$	$F_{GRM}$	1	7	0.186	<0.001	7	100
$F_{PED}$	$F_{GRM0.5}$	4	26	0.046	0.005	21	81
$F_{PED}$	HOM	4	18	0.238	0.029	15	83
$F_{ROH}$	$F_{GRM}$	1	7	0.307	0.238	4	57
$F_{ROH}$	$F_{GRM0.5}$	5	40	0.037	0.036	15	38
$F_{ROH}$	HOM	4	34	0.158	0.170	19	56
$F_{GRM}$	$F_{GRM0.5}$	1	3	0.046	0.002	3	100
$F_{GRM}$	HOM	2	14	0.071	0.009	4	29
$F_{GRM0.5}$	HOM	1	20	0.280	0.170	10	50

*P*1 = *p*-value of measure 1; *P*2 = *p*-value of measure 2.

## Data Availability

All data and results supporting this study are provided in the main text, Appendix B and Appendix A.

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
