# Peer review of "How Depressing Is Inbreeding? A Meta-Analysis of 30 Years of Research on the Effects of Inbreeding in Livestock"

_genes, 2021, doi:10.3390/genes12060926_

Round 1
Reviewer 1 Report
Manuscript title: How depressing is inbreeding? A meta-analysis of 30 years research on the effects of inbreeding in livestock
Line 2: Please add "of" to the title (30 years of research)
Line 20: Strange to mention production traits with morphology in the same category
Introduction: The Introduction part explains the background of the study properly. The reasons for the study are clear, the authors aimed to compare the inbreeding depression values of different traits and trait categories of different datasets.
The methods are clear and well-described however, there are some remarks regarding the chapter.
Line 115 and TableS1: "Other traits" category contains behavioral traits and performance traits resulting in a great variation of trait means. Maybe this categorization should be reconsidered.
Line 145: Authors used Tukey's method for comparing EMMs. For non-equal sample sizes, the Tukey-Kramer method would be correct.
Line 159: above mentioned
Results
Line 378: Authors should describe the final model here. What is a general population effect?
Supplementary files: InTableS1 variable Nanimals have the value of ?: is it equal to NA or is it a different value?
Reviewer 2 Report
The authors present a comprehensive meta-analysis on inbreeding depression for livestock species, which will be certainly of interest for many researchers. The presentation is rather clear and the analyses are well explained, with a detailed justification of the decisions made regarding the filtering of data. I think this paper will be a good contribution to the field and may be highly cited.
I have a few comments which may help to improve the paper.
(1) The analysis shows that ID for reproduction/survival traits is of similar magnitude as that from other traits, or even smaller in the particular case when b_s is used (ID as a percentage of trait´s SD). There are a couple of issues which can be related with this observation (this point and the next one). Reproductive traits are known to have generally much more phenotypic variation from environmental sources than other traits, so a possible scale factor (regarding trait´s SD) might be playing a role when b_s values are compared. This may be analogous to the observation that fitness traits tend to have lower heritability than morphological traits, while the amount of additive genetic variance of fitness traits is not lower, but often larger, than that of morphological traits. This is shown when the coefficient of additive genetic variation (CVA) is compared instead of the heritability. I wonder if the observation that b_s for reproduction/survival traits is lower than that for other traits can be explained by the same effect, and some sort of correction by trait variation (CV) may be applied.
(2) Because populations of livestock species generally have relatively small effective population sizes (although they may have very large census sizes), the authors may consider to discuss the possibility of genetic purging being a factor for reducing the ID for fitness traits. Purging of the deleterious inbreeding load can be very efficient for moderately large (say Ne = 50 – 100) effective population sizes (see e.g. López-Cortegano et al. 2016. Evolution 70: 1856–1870, and references therein). This may be added to the explanation given by the authors in terms of artificial selection (lines 344-348).
(3) Line 152: It is unclear if the values of ID from F_GRM considered are referring to VanRaden1, VanRaden2 or Yang, or to all of them mixed, which is what I guess.
(4) Line 411: Caballero et al. simulations did not consider simple SNP homozygosity (HOM) directly, but analysed the estimator F_hom, proposed by Li & Horvitz (1951), which has a correlation of 1 with HOM. Thus, both are equivalent, except that the first makes a scaling correction to estimate IBD if base population frequencies are known.
(5) I have missed some discussion in relation with the observed correlations presented in Figures 3 and 4. What can be learned/concluded from these overall correlations? In addition, although these refer to ID estimates, they are surely related with the different estimators of F. There are many empirical and simulatation results regarding the correlations between these estimators, which (at least some of them) can be compared/discussed with the meta-analysis values.
(6) Line 434 and onwards. Within the list of recommendations given by the authors for future studies to estimate ID, I would suggest to add that, when available, estimates of the effective population size should also be provided. The reason is that, as mentioned above, the amount of genetic purging of the inbreeding load, and thus the ID for traits with deleterious genetic variation, critically depends on the effective size.
(7) Line 505: A minor issue. The equation of VR2 was actually proposed by Leutenegger (2003) and Amin (2007), as mentioned by VanRaden (2008; page 4416).
(8) Can be something said about differences (or not) of ID between species?
(9) Table 2: Can you give more insight into the reasons for the higher P-value of molecular F estimates with respect to those from Fped? Will this comparison depend on the amount/density/reliability of markers and sample sizes?
